# Extracellular Vesicle microRNA: A Promising Biomarker and Therapeutic Target for Respiratory Diseases

**DOI:** 10.3390/ijms25179147

**Published:** 2024-08-23

**Authors:** Jiaxi Lv, Xianzhi Xiong

**Affiliations:** 1Department of Pulmonary and Critical Care Medicine, Union Hospital, Tongji Medical College, Huazhong University of Science and Technology, 1277 Jiefang Avenue, Wuhan 430022, China; lvjiaxi@hust.edu.cn; 2Department of Respiratory and Critical Care Medicine, Union Hospital, Tongji Medical College, Huazhong University of Science and Technology, 1277 Jiefang Avenue, Wuhan 430022, China

**Keywords:** respiratory disease, extracellular vesicle, microRNA, chronic obstructive pulmonary disease, asthma, lung cancer, coronavirus pneumonia

## Abstract

Respiratory diseases, including chronic obstructive pulmonary disease (COPD), asthma, lung cancer, and coronavirus pneumonia, present a major global health challenge. Current diagnostic and therapeutic options for these diseases are limited, necessitating the urgent development of novel biomarkers and therapeutic strategies. In recent years, microRNAs (miRNAs) within extracellular vesicles (EVs) have received considerable attention due to their crucial role in intercellular communication and disease progression. EVs are membrane-bound structures released by cells into the extracellular environment, encapsulating a variety of biomolecules such as DNA, RNA, lipids, and proteins. Specifically, miRNAs within EVs, known as EV-miRNAs, facilitate intercellular communication by regulating gene expression. The expression levels of these miRNAs can reflect distinct disease states and significantly influence immune cell function, chronic airway inflammation, airway remodeling, cell proliferation, angiogenesis, epithelial-mesenchymal transition, and other pathological processes. Consequently, EV-miRNAs have a profound impact on the onset, progression, and therapeutic responses of respiratory diseases, with great potential for disease management. Synthesizing the current understanding of EV-miRNAs in respiratory diseases such as COPD, asthma, lung cancer, and novel coronavirus pneumonia, this review aims to explore the potential of EV-miRNAs as biomarkers and therapeutic targets and examine their prospects in the diagnosis and treatment of these respiratory diseases.

## 1. Introduction

Extracellular vesicles (EVs) are membrane-bound vesicles containing proteins, lipids, DNA, RNA, and other biomolecules that are released by cells into the extracellular environment, ultimately interacting with target cells [1]. MicroRNAs (miRNAs) are regulatory non-coding RNAs, typically 19 to 22 nucleotides in length, that regulate gene expression by binding to the 3′ untranslated region (3′ UTR) of target mRNAs, resulting in mRNA degradation or inhibition of translation [2]. Encapsulated within EVs, miRNAs can be transported over long distances and exhibit high stability due to their protection from enzymatic degradation in the extracellular environment [3]. These miRNAs play crucial roles in numerous physiological and pathological processes. Investigating the mechanisms by which EV-miRNAs influence disease progression, as well as their differential expression in various disease states, provides novel insights for early diagnosis, liquid biopsies, and the identification of new therapeutic targets. This field holds significant promise for the development of precision treatment and personalized medicine, thereby receiving widespread attention and presenting substantial research potential.

Respiratory diseases encompass diverse conditions, including ventilatory disorders (obstructive and restrictive), lung cancer, and pulmonary infectious diseases. Among these, chronic obstructive pulmonary disease (COPD) and asthma are the most prevalent chronic respiratory diseases, significantly impairing patients’ work capacity and quality of life and imposing a substantial socioeconomic burden [4,5]. Lung cancer, the leading cause of cancer incidence and mortality [6], also represents a critical threat to human health, particularly in developing countries with limited medical resources and poor capabilities in cancer screening. Coronavirus pneumonia (COVID-19) has rapidly escalated into a global public health crisis since its outbreak in 2019 [7]. The pandemic has strained medical resources, severely disrupted economic activities and social order, and profoundly altered people’s lifestyles [8].

However, early diagnosis of these diseases remains challenging, and current treatment methods are relatively rudimentary. The diagnosis of COPD and asthma primarily relies on symptom assessment and lung function tests, which often fail to detect the diseases in their early stages, resulting in delayed treatment. It is estimated that approximately 70–80% of adults with COPD remain undiagnosed [9]. Similarly, the early diagnosis of lung cancer is problematic, with many patients being undiagnosed until advanced stages when symptoms manifest, thereby resulting in suboptimal treatment [10]. The rapid spread and mutation of COVID-19 have complicated disease control and vaccine development [11], necessitating rapid virus detection and precise treatment strategies.

Meanwhile, aging population exacerbates the incidence and mortality rates of these respiratory diseases, which are expected to continue rising globally [12]. Moreover, the therapeutic outcomes for these diseases are influenced by individual differences, including genetics, age, gender, lifestyle, and personal health conditions. This variability necessitates highly personalized treatment approaches, increasing the complexity of therapy. In this context, research on early diagnosis, effective treatment, and prevention strategies for respiratory diseases is critically important. Identifying new biomarkers and therapeutic targets is essential for advancing precision medicine through improved early diagnosis and efficacy assessment. This area of research remains a significant focus, aiming to improve the management and outcomes of respiratory diseases.

Due to the high incidence or mortality rates of COPD, asthma, lung cancer, and COVID-19 and the significant importance of early diagnosis for guiding treatment and prognosis, this review will focus on the current state of research on EV-miRNA in these four diseases. Accumulating studies reveal the important role and promising future of EV-microRNAs in these four diseases. This review aims to elucidate the role of EV-microRNAs in COPD, asthma, lung cancer, and coronavirus pneumonia, with the hope of inspiring advancements in their clinical management.

## 2. EV-microRNA

Extracellular vesicles (EVs) are a collective term for various membrane-structured vesicles released by cells. Based on their size, biological origin, and other characteristics, EVs can be broadly categorized into exosomes, microvesicles, and apoptotic bodies [13]. Table 1 outlines the key characteristics of these three types of EVs:

Exosomes, the most extensively studied type of extracellular vesicles, typically range from 30 to 150 nm in diameter. Generally, during endocytosis, the plasma membrane undergoes invagination and forms early endosomes. These endosomes interact with the Golgi apparatus, endoplasmic reticulum, mitochondria, phagosomes, RNA granules, and micronuclei to sort proteins, lipids, RNA, and DNA into the endosomes, forming mature endosomes. Mature endosomes then form multivesicular bodies (MVBs), which are secreted outside the cell as exosomes [14]. Once exosomes are taken up by target cells, the biomolecules within them can be released into the cytoplasm, influencing gene expression, signal transduction, cell differentiation, proliferation, and apoptosis. This intercellular communication function of exosomes is crucial in numerous biological processes, including immune modulation, tissue repair, regulation of the tumor microenvironment, and the development of diseases. Almost all human cells can produce exosomes, and the cargo biomolecules and functions of exosomes produced by different parental cells vary significantly [15].

Microvesicles, with a diameter of approximately 100–1000 nanometers, are formed through a process closely related to the dynamic changes of the plasma membrane. They are generated by budding from the plasma membrane and subsequent shedding, a process independent of endocytosis and distinct from the mechanism of exosome formation. Microvesicles contain various substances such as proteins, lipids, nucleic acids (including mRNA and miRNA), and fragments of organelles. Like exosomes, microvesicles can mediate intercellular communication, thereby influencing a variety of biological functions [16].

Apoptotic bodies are a type of extracellular vesicle formed during the process of apoptosis, exhibiting a wide range of sizes, from tens of nanometers to several micrometers. They are the final products of apoptotic cell death, formed by the budding and subsequent shedding of the cell membrane. Signaling molecules on the surface of apoptotic bodies, such as phosphatidylserine, facilitate the interaction between apoptotic bodies and phagocytic cells, thereby initiating phagocytosis. This process is crucial for the recognition and clearance of apoptotic cells by phagocytes, maintaining tissue homeostasis, and reducing inflammation [17].

Common methods for the extraction and purification of EVs include Ultracentrifugation, size-based separation, Immunoaffinity Capture, Polymer Precipitation, Microfluidics, and the Kit Method. The principles, advantages, and disadvantages of these isolation methods [13] are detailed in Table 2. Currently, Ultracentrifugation remains the gold standard for the separation of EVs [18]. Ultracentrifugation encompasses Differential Ultracentrifugation and Density Gradient Ultracentrifugation. In Density Gradient Ultracentrifugation, density medium is added during the centrifugation process, allowing EVs to precipitate at specific density layers. Despite the long operation time, complex procedures, and possible impact on the biological activity of EVs, Ultracentrifugation is still widely used due to its ability to process large sample volumes, applicability to various biological samples, and the relative purity of EVs obtained. Ultracentrifugation can be combined with other methods to enhance the purity and recovery rate of EVs [18].

MiRNAs are small, non-coding RNA molecules approximately 21 nucleotides in length [19]. Most miRNA gene sequences are located within the introns or exons of non-coding RNA or within the introns of precursor mRNA (pre-mRNA). Generally, miRNA genes are transcribed into primary miRNA (pri-miRNA) by RNA polymerase II (Pol II), which contains stem-loop structures. Within the nucleus, pri-miRNA is cleaved into precursor miRNA (pre-miRNA) by the microprocessor complex composed of Drosha and DGCR8. The pre-miRNA is then transported to the cytoplasm by Exportin 5 (XPO5), where it is further cleaved by the Dicer enzyme into mature double-stranded RNA (dsRNA), ultimately forming mature miRNA capable of guiding the RNA-induced silencing complex (RISC) [2,20]. The core component of RISC is the Argonaute (Ago) protein, which possesses endonuclease activity and can bind to complementary sequences on the 3′ untranslated region (3′UTR) of target mRNA, leading to mRNA degradation or inhibition of its translation, thereby reducing the production of specific proteins [21]. This mechanism is known as post-transcriptional regulation. MiRNAs can regulate multiple genes, while a single gene may also be regulated by multiple miRNAs, forming a complex gene regulatory network. This network plays a key role in various biological processes such as cell development, differentiation, metabolism, and stress responses [2].

The main methods for the quantification of microRNAs include Northern Blot, quantitative real-time polymerase chain reaction (qRT-PCR), microarray, and Next-Generation Sequencing (NGS) [2,22]. The principles, advantages, and disadvantages of these methods are detailed in Table 3 below. Among them, qPCR has become the preferred method for miRNA detection due to its high sensitivity, high throughput, speed, cost-effectiveness, and technical maturity [22].

MicroRNAs can be categorized based on their location as either intracellular or extracellular. Intracellular microRNAs primarily function within the cell, interacting with the genome in the nucleus to regulate gene expression. In contrast, extracellular microRNAs, which avoid degradation by ribonucleases (RNases) in body fluids, can be further subdivided into EVs-free circulating microRNAs and EV-microRNAs [3]. EVs-free circulating microRNAs can bind to High-Density Lipoprotein (HDL) or Low-Density Lipoprotein (LDL), or ribonucleoproteins such as AGO2 or NPM1, thereby being protected from degradation [23]. EV-microRNAs are microRNAs sorted into EVs, which are produced within the parental cell and selectively packaged into EVs for secretion [24]. This process involves various RNA-binding proteins (RBPs), such as hnRNPA2B1, Argonaute-2 (Ago2), Y-Box binding protein 1 (YBX-1), MEX3C, and La protein, which facilitate the entry of microRNAs into EVs by binding to specific sequences of the microRNAs. In addition to RBPs, some membrane proteins, such as Caveolin-1 (Cav-1) and neutral sphingomyelinase 2 (nSMase2), also participate in the biogenesis of EVs and the selective loading of miRNAs. This process can be regulated by signaling pathways [24]. Encapsulated within the membranous confines of EVs, EV-microRNAs exhibit exceptional stability, enabling their prolonged presence in bodily fluids and facilitating their transportation to distant cellular destinations over protracted durations. Upon discharge into the extracellular milieu, EVs traverse through the bloodstream or alternate bodily fluid conduits, effectively modulating the gene expression of recipient cells. These entities serve as pivotal mediators of intercellular communication, exerting profound influences on both physiological homeostasis and pathological states.

## 3. EV-microRNAs and Respiratory Diseases

EV-microRNAs can be secreted by various types of cells, including epithelial cells, immune cells, endothelial cells, mesenchymal stem cells, and tumor cells. They are widely present in multiple biofluids and can be detected through fluid samples such as blood [25], urine, and bronchoalveolar lavage fluid while maintaining their stability. The collection of such samples entails relatively non-invasive procedures, thereby facilitating extensive screening initiatives and ongoing monitoring endeavors. Additionally, certain diseases can lead to the dysregulation of EV-microRNA content. The expression profile of EV-microRNAs may closely correlate with an individual’s genetic background and disease state, thereby informing personalized medicine and precision therapy [24]. Due to these attributes, EV-microRNAs are considered potential biomarkers, garnering significant attention from researchers focused on respiratory diseases [26]. An increasing number of studies have demonstrated that EV-microRNAs play important role in the pathogenesis of COPD, asthma, lung cancer and COVID19 as detailed in Table 4. The differential expression levels of EV-microRNAs can reflect pathological conditions. These changes may be linked to mechanisms such as oxidative stress, apoptosis and regeneration, inflammation and immunity, and cellular signaling. Consequently, EV-microRNAs hold potential as biomarkers for the diagnosis and assessment of the severity of these diseases [27,28,29,30].

### 3.1. EVs and EV-microRNAs Maintain Pulmonary Homeostasis

Various cell types in the lungs secrete EVs that carry specific microRNAs, proteins, and other bioactive molecules, playing a crucial role in maintaining pulmonary homeostasis. EVs derived from bronchial epithelial cells contain surface-associated mucins (MUC)-1, MUC-4, and MUC-16, which are essential for innate mucosal defense [45,46]. Meanwhile, EVs originating from alveolar epithelial cells can transport suppressor of cytokine signaling (SOCS) proteins and miR-223, which are involved in modulating inflammatory signals [47,48]. Endothelial cell-derived EVs can inhibit monocyte activation by delivering anti-inflammatory miR-10a, which may help protect the lungs from inflammatory damage [49]. Immune cell-secreted EVs also play a significant role in the pulmonary microenvironment; for instance, dendritic cell-derived EVs are involved in the activation of T cells, while mast cell-derived EVs contribute to the production of the antibody IgE [50].

### 3.2. EV-microRNAs and COPD

Chronic respiratory disease (CRD) ranks as the third-leading cause of death worldwide, with chronic obstructive pulmonary disease (COPD) being a major contributor to the prevalence and mortality rates associated with CRD. In 2019, there were 212.3 (200.4–225.1) million prevalent cases of COPD globally, with 16.2 (15.2–17.2) million new cases and 3.3 (2.9–3.6) million deaths [5]. COPD severely reduces patients’ quality of life, compromises work capacity, and imposes a substantial economic burden on society. Thus, early diagnosis, effective treatment, and long-term management of COPD are of great importance.

COPD is defined as a group of heterogeneous chronic respiratory diseases characterized by persistent airflow obstruction. The development of COPD results from the interplay between genetic and environmental factors. Smoking is the primary risk factor for COPD, with various components in cigarette smoke being closely linked to its progression [51]. The pathogenesis of COPD involves several mechanisms, including inflammation, an imbalance between protease and antiprotease activities, oxidative stress, cellular damage, aging, and apoptosis [52,53,54,55,56,57,58,59]. These mechanisms collectively contribute to airway remodeling, hypersecretion, and the destruction of alveolar structures, resulting in clinical manifestations such as chronic bronchitis and emphysema.

Currently, the diagnosis of COPD primarily relies on spirometry [51]. By the time COPD patients receive diagnosis, they are often at an advanced stage with severe symptoms of airflow limitation. However, despite advancements in medical care, the treatment of COPD remains primarily symptomatic, focusing on slowing disease progression, alleviating symptoms, and reducing the occurrence of severe complications. These limitations in both diagnosis and treatment stem in part from our incomplete understanding of the mechanisms underlying COPD development and progression. Investigating the relationship between EV-microRNAs and COPD holds promise for providing new insights and opportunities to comprehend the intricate mechanisms of COPD and identify potential biomarkers for its early diagnosis and treatment.

Smoking can impact intercellular communication between epithelial cells and fibroblasts. Alterations in EV-microRNAs originating from airway epithelial cells may foster airway remodeling by facilitating the differentiation of myofibroblasts, thus contributing to the onset and advancement of COPD. In 2015, Fujita et al. [31] discovered that following exposure to cigarette smoke extract (CSE), the expression of miR-210 in EVs derived from human bronchial epithelial cells (HBECs) was upregulated. MiR-210 can regulate the autophagy process by directly targeting the autophagy-related protein 7 (ATG7), and silencing the ATG7 gene in lung fibroblasts (LFs) leads to their differentiation into myofibroblasts. The authors concluded that the upregulation of miR-210 in EVs from HBECs after exposure to CSE may control the differentiation of myofibroblasts by regulating the autophagy process, thereby contributing to airway remodeling. In 2018, Xu et al. [32] observed that exposure to CSE led to upregulation in the expression of miR-21 within EVs derived from HBECs. MiR-21 is known to target the Von Hippel-Lindau protein (pVHL), and the subsequent downregulation of pVHL expression can elevate the levels of hypoxia-inducible factor 1α (HIF-1α), as well as the protein levels of α-smooth muscle actin (α-SMA) and collagen type I. These changes collectively promote the differentiation of myofibroblasts. They also reported that administration of a miR-21 inhibitor could prevent cigarette smoke (CS)-induced airway remodeling in a mouse model. Furthermore, they observed elevated levels of exosomal miR-21 in the serum of smokers and patients with COPD, which were negatively correlated with the FEV1/FVC ratio (the ratio of forced expiratory volume in one second to forced vital capacity).

Smoking can also influence intercellular communication between airway epithelial cells and macrophages, thereby contributing to the development of COPD through the modulation of macrophage function. In 2022, Xia et al. [33] observed that exposure to CS induces heightened levels of METTL3, miR-93, and MMPs in mouse lung tissue, concomitant with the onset of emphysema. This phenomenon may stem from METTL3′s facilitation of pri-miR-93 maturation via enhanced m6A modification, consequently elevating miR-93 levels. Notably, miR-93, transported by EVs secreted by airway epithelial cells, targets dual-specificity phosphatase 2 (DUSP2) in macrophages, thereby activating the JNK pathway. This activation leads to increased secretion of matrix metalloproteinases (MMPs), particularly MMP9 and MMP12, by macrophages, culminating in elastin degradation and the subsequent manifestation of emphysema in COPD. Furthermore, the authors reported a negative correlation between the level of miR-93 in EVs and the FEV1/FVC ratio in patients with COPD.

In addition to the previously discussed intercellular communication between epithelial cells and macrophages, smoking can also impact the communication between vascular endothelial cells and macrophages. In 2016, Serban et al. [34] conducted research on the effects of CSE exposure on the release of circulating endothelial microparticles (EMPs) and their functional characteristics. They discovered that CSE exposure increased the levels of EMPs in the plasma of both humans and mice. These EMPs, primarily exosomes, were enriched with specific microRNAs, including let-7d, miR-191, miR-126, and miR-125a. The researchers observed that when plasma monocyte-derived macrophages engulfed these EMPs, their phagocytic function diminished. They inferred that the altered expression levels of these microRNAs might impair macrophage phagocytic function by reducing cell motility and affecting cytokine secretion. This study suggests that EV-microRNAs released by endothelial cells play a significant role in the pathological processes of COPD, particularly in endothelial damage and inflammatory responses, thereby offering new potential targets for the diagnosis and treatment of COPD.

Some studies have shown that EV-microRNAs can assist in determining the severity of COPD. In 2020, Carpi et al. [60] analyzed the expression of muscle-specific microRNAs in extracellular vesicles (EVs) derived from the plasma of COPD patients and found that miR-206, miR-133a-5p, and miR-133a-3p were upregulated in these patients. They inferred that these microRNAs might serve as biomarkers for muscle dysfunction in COPD patients and could help distinguish different degrees of COPD severity.

Acute exacerbation of COPD (AECOPD) represents a clinical deterioration in COPD patients, typically characterized by increased difficulty in breathing, coughing, increased sputum volume, or changes in the nature of the sputum. AECOPD is a major cause of hospitalization and mortality among COPD patients, significantly impacting their quality of life and long-term prognosis [51]. Identifying biomarkers for AECOPD is crucial for enhancing the diagnosis, treatment, and management of COPD, thereby improving patient quality of life and survival rates. EV-microRNAs have the potential to serve as biomarkers, aiding in the early identification of AECOPD. In 2022, Farrell et al. [61] investigated the expression of plasma EV-microRNAs and Receiver Operating Characteristic (ROC) curves in two cohorts: AECOPD and stable COPD (SCOPD). They found significant dysregulation in the expression of five microRNAs between the AECOPD and SCOPD groups. Among these, miR-223-3p had the highest Area Under the Curve (AUC) at 0.755. When combined with miR-92b-3p, miR-374a-5p, and miR-106b-3p, the AUC increased to 0.820, indicating a higher diagnostic accuracy for distinguishing AECOPD from SCOPD. In 2023, Wang et al. [62] studied the expression of serum exosomal microRNAs in normal controls (NC), patients with SCOPD, and patients with AECOPD. They found that exosomal miR-1258 outperformed other microRNAs in distinguishing AECOPD from SCOPD. Additionally, the diagnostic model combining exosomal miR-1258 and neutrophil count (NEU) demonstrated the highest accuracy in diagnosing AECOPD. Both studies emphasize the potential of EV-microRNAs as diagnostic biomarkers for AECOPD and propose further research to validate these findings in larger cohorts of patients.

Additionally, some studies have suggested the potential of EV-microRNAs in distinguishing COPD from other diseases. In 2021, Farrell et al. [63] examined plasma EV-microRNA expression in healthy non-smokers, healthy smokers, lung cancer patients, and patients with SCOPD. They discovered that certain microRNAs can assist in differentiating lung cancer from healthy smokers and patients with SCOPD. Specifically, they identified 14 microRNAs with significantly different expression levels between lung cancer patients and those with stable COPD. Among these, miR-27a-3p had the highest area under the receiver operating characteristic curve (AUC = 0.803). When combined with miR-106b-3p and miR-361-5p, the AUC increased to 0.870, indicating enhanced diagnostic accuracy. In 2022, Burke et al. [64] found that some differentially expressed EV-microRNAs in the bronchoalveolar lavage fluid (BALF) could indicate inflammatory endotypes such as neutrophilic and eosinophilic COPD. These findings may be helpful for treatment stratification, allowing for more personalized and targeted therapeutic approaches.

In summary, EV-microRNAs have the potential to serve as biomarkers for the diagnosis and assessment of disease status and severity in COPD. Current research has primarily focused on the impact of smoking-induced changes in EV-microRNAs on COPD. In recent years, on the other hand, non-smoking causes of COPD have also received attention [65]. The relationship between changes in EV-microRNAs and the development of COPD due to non-smoking factors such as air pollution and occupational exposure may become a significant direction for future research. In addition, accumulating studies suggest that COPD can be regarded as a pulmonary manifestation of a systemic disease [54,66,67]. Studying the changes in EV-microRNA in the comorbid condition of COPD and other diseases, such as cardiovascular diseases and diabetes, may provide a new perspective for further understanding. Currently, the diagnosis of COPD requires confirmation through pulmonary function tests; consequently, some individuals with chronic respiratory symptoms or radiological manifestations may not be diagnosed with COPD or receive treatments. However, there is a viewpoint that chronic damage to the respiratory tract has already begun at this “pseudo-normal” stage, where the patient’s age-pulmonary function curve may exhibit a subnormal pattern [54]. Although this theory still needs confirmation, studying the changes in EV-microRNA levels among “pseudo-normal” individuals, healthy populations, and COPD patients may offer more insights into the management of COPD.

### 3.3. EV-microRNAs and Asthma

Asthma is one of the major contributors to the prevalence of chronic respiratory diseases worldwide. Studies have shown that in 2019, approximately 262 million people suffered from asthma globally, with about 37 million new cases, imposing a significant burden on health and the economy [5]. Although the prevalence of asthma has declined since 1990 [5], many challenges remain in its diagnosis and treatment. Many asthma patients remain undiagnosed and untreated. According to a study in China, it is estimated that about 45.7 million Chinese adults have asthma, but only 28.8% have been diagnosed, 23.4% have undergone pulmonary function testing, and 5.6% have received inhaled corticosteroid treatment [68]. Therefore, early diagnosis, treatment, and management of asthma remain important issues in the field of global public health.

Asthma is a heterogeneous group of chronic respiratory diseases characterized by recurrent episodes of intermittent bronchospasm and reversible airway obstruction [69]. The etiology of asthma is related to various factors, including genetic predisposition, environmental exposures, and individual allergic constitution [70]. The mechanisms of asthma can be broadly summarized as immune-inflammatory responses [71,72], airway hyperresponsiveness (AHR), and neural mechanisms [73]. As asthma progresses, patients may undergo a series of structural alterations in the airways, a process referred to as airway remodeling [74].

The diagnosis of asthma relies on lung function tests such as bronchial dilation tests. Treatments can be broadly categorized into quick-relief (rescue) medications and long-term control medications, including anti-inflammatory drugs, bronchodilators, and targeted therapy such as biologics [75]. Treatments for asthma typically aim to manage symptoms, reduce the frequency of asthma attacks, maintain normal lung function, and improve quality of life, emphasizing the importance of personalized treatment and precision medicine [76]. Therefore, the search for better biomarkers has become the focus of researchers. Accumulating studies have found that EV-microRNAs play an important role in the development of asthma and may serve as biomarkers, bringing new possibilities for the diagnosis and treatment of the disease.

Asthma has multiple causes, including environmental factors such as indoor allergens (dust mites and pets), outdoor allergens (pollen and grasses), and non-allergic factors such as air pollution. Respiratory infections, particularly viral infections, are also common triggers for asthma attacks [77]. Some studies have examined the impact of these asthma causes on EV-microRNA. In 2016, Gutierrez et al. [78] characterized the baseline airway-secreted EV-microRNAs in children and examined the changes during rhinoviral (RV) infection. They found that hsa-miR-155 was the primary change relative to the baseline microRNA group during RV infection. They concluded that miR-155 may inhibit the replication of rhinoviruses and other viruses, regulate immune cell function, and play a role in the Th2 immune response. In 2017, Gon et al. [79] investigated changes in airway-secreted EVs (AEVs) and their microRNAs in airway inflammation induced by house dust mite (HDM) allergens. They found that HDM exposure led to a significant increase in EVs in mouse BALF, along with noticeable changes in EV-microRNAs in both mouse BALF and lung tissues. The expression changes of microRNAs in AEVs showed a negative correlation with those observed in the lung tissues. Based on this observation, they speculated that the secretion of microRNAs into the airway lumen via EVs might facilitate allergic airway inflammation, potentially contributing to one of the pathogenic mechanisms underlying asthma. To explore further, they employed the sphingomyelinase inhibitor GW4869, known to decrease the release of EVs. Subsequently, they administered GW4869 and noted a reduction in the number of AEVs in the BALF post-treatment. This reduction was accompanied by decreased levels of Th2 cytokines and eosinophil counts in the BALF, as well as a decrease in eosinophil accumulation in the airway wall and mucosa. These findings provide evidence supporting the involvement of EVs in allergic airway inflammation and suggest potential therapeutic targets for the development of new treatment strategies. In 2022, Zheng et al. [80] investigated the impact of PM2.5 on asthma development through intercellular communication. They discovered that EVs secreted by PM2.5-treated HBECs (PM2.5-EVs) could induce cytotoxicity in HBECs and enhance the contractility of sensitive human bronchial smooth muscle cells. Notably, they observed elevated levels of let-7i-5p in PM2.5-EVs and in the plasma of asthmatic patients. Furthermore, they identified a correlation between the expression of let-7i-5p in PM2.5-EVs and PM2.5 exposure in children with asthma. Their research suggests that let-7i-5p in PM2.5-EVs activates the MAPK signaling pathway by inhibiting the dual specificity phosphatase 1 (DUSP1) gene, thereby exacerbating PM2.5-induced asthma. These studies offer insights into the connection between etiology and the onset of the disease from a novel perspective, and they present new possibilities for diagnosis and treatment.

Several studies have investigated the relationship between immune cells and EV-microRNAs in asthma. In 2019, Pua et al. [81] examined the expression of EVs and miRNA in bronchoalveolar lavage fluid (BALF) from mice before and after lung allergen challenge. They observed that prior to the induction of allergic inflammation, 80% of EVs originated from epithelial cells. However, following inflammation, there was a significant increase of more than 2-fold in EVs derived from immune cells, accompanied by the presence of immune cell-specific miRNAs such as miR-223 and miR-142a. These findings led to the conclusion that during allergic inflammation, immune cells release EVs and EV-miRNAs, thereby influencing the local extracellular environment. In 2019, Zhuansun et al. [35] discovered that EV miR-1470 derived from mesenchymal stem cells (MSCs) could enhance the proportion of CD4^+^CD25^+^FOXP3^+^ regulatory T cells (Tregs) in asthma by stimulating the expression of P27KIP1. They found that miR-1470 facilitated the upregulation of P27KIP1 by directly targeting the 3′ region of c-Jun mRNA. Through the inhibition of P27KIP1 via siRNA, they observed a significant reduction in both the overexpression of miR-1470 and the proportion of CD4^+^CD25^+^FOXP3^+^ Tregs. They therefore concluded that miR-1470 induces the differentiation of CD4^+^CD25^+^FOXP3^+^ Tregs through P27KIP1. In 2021, Li et al. [36] investigated M2 macrophage-derived exosomes (M2Φ-Exos) in asthma. They proposed that M2Φ-Exos could potentially alleviate the advancement of asthma by transporting miR-370, consequently suppressing the expression of FGF1 and the MAPK/STAT1 signaling pathway.

In recent years, there has been growing interest in exploring the potential of EV-microRNAs as biomarkers for asthma. In 2013, Sinha et al. [82] highlighted the potential of EV-microRNAs in exhaled breath condensate (EBC) as biomarkers for lung diseases. Despite the noted high stability of microRNAs in EBC, the authors have emphasized the necessity for additional testing to ascertain their clinical utility. Meanwhile, the tissue origin of exosomes in EBC remains unclear. In 2013, Levänen et al. [83] observed significant differences in 24 microRNAs in the BALF of asthma patients compared to the healthy control group. Among these, 16 microRNAs, including members of the let-7 and microRNA-200 families, showed promise in identifying mildly asymptomatic asthma. These EV-microRNAs likely contribute to bronchial hyperresponsiveness and the inflammatory response in asthma. In 2016, Panganiban et al. [84] conducted research analyzing circulating microRNAs among asthma patients, patients with allergic rhinitis but without asthma, and healthy controls. They observed that the expression of circulating miRNAs in asthma patients could be categorized into two subtypes, representing high and low eosinophilic states. Additionally, they identified circulating miR-125b, miR-16, miR-299-5p, miR-126, miR-206, and miR-133b as potential biomarkers for predicting allergic diseases and asthmatic conditions. In 2019, Zhao et al. [85] reported elevated expression levels of miR-126 in the serum exosomes of patients with allergic asthma. Similar findings were observed in the lung tissue of asthmatic mice, suggesting the involvement of exosomal miR-126 in the pathogenesis of bronchial asthma. Additionally, in the same year, they found that the expression level of miR-125b in serum exosomes exhibited high diagnostic efficacy and may serve as a non-invasive diagnostic marker for asthma severity [86]. In 2021, Rostami Hir et al. [87] investigated the expression levels of plasma EV-microRNAs in patients with allergic asthma in comparison to healthy controls. They observed significant elevations in the levels of miR-223 and miR-21 among patients with moderate asthma as opposed to the healthy control group. They concluded that these microRNAs could potentially serve as adjunctive diagnostic biomarkers or therapeutic targets. In 2023, Piera Soccio et al. [88] reported on differences in circulating microRNAs and EV-microRNAs in patients with mild to moderate and severe asthma. They observed that, compared to the healthy control group, circulating miR-21 levels were significantly elevated in patients with severe asthma, while circulating let-7a levels were increased in patients with mild to moderate asthma. Additionally, exosomal miR-223, miR-21, and let-7a levels were also elevated in patients with severe asthma. These findings may offer new possibilities for the early diagnosis and phenotyping of asthma.

Additionally, some studies have focused on the potential significance of EVs and EV-microRNAs in the treatment of asthma. In 2023, Soyoon Sim et al. [89] reported that Micrococcus luteus-derived extracellular vesicles (MlEVs) may alleviate neutrophilic asthma (NA) by regulating microRNA in airway epithelial cells (AECs). They concluded that MlEVs upregulated the expression of hsa-miR-4517 in AECs, a microRNA capable of inhibiting IL-1β production by monocytes and suppressing the activation of ILC3 and the recruitment of neutrophils. In 2023, Xiaowei Xu et al. [90] reported that the inhalation of hypoxia-cultured human umbilical cord mesenchymal stem cell-derived extracellular vesicles (Hypo-EVs) might suppress asthmatic airway inflammation and remodeling. This study suggested that the inhalation of Hypo-EVs effectively reduced airway inflammation, decreased the levels of inflammatory factors, and alleviated airway remodeling. The authors concluded that inhaling Hypo-EVs might become an effective, non-invasive treatment for asthma. Another study went further. In 2024, Wei Tu et al. [91] demonstrated in a mouse model that engineered EV-microRNAs have the potential to alleviate allergic airway inflammation. They loaded miR-511-3p mimics into mannose-modified EVs (Man-EV-miR-511-3p) and administered them via airway inhalation to miR-511-3p knockout mice exposed to cockroach allergens. They observed that the absence of miR-511-3p led to exacerbated airway hyperresponsiveness and Th2-associated allergic inflammation in mice. They concluded that Man-EV-miR-511-3p could penetrate the airway mucus barrier, act on lung macrophages, directly bind and activate the C3 gene, polarize macrophages towards the M2 phenotype, and thus suppress allergic airway inflammation. This study not only revealed the role of miR-511-3p in the pathogenesis of asthma but also showed the potential of RNA nanotechnology and exosomes as carriers for microRNA delivery, providing a new strategy for the treatment of respiratory diseases.

In summary, EV-microRNAs hold significant value in the development, diagnosis, and treatment of asthma. One key aspect of asthma management is reducing acute exacerbations, and the relationship between EV-microRNAs and acute asthma attacks warrants further in-depth study. Besides, the role of EV-microRNAs in different phenotypes and endotypes of asthma also requires further investigation. There is also a lack of research on the relationship between EV-microRNAs and specific disease states, such as COPD-asthma overlap syndrome, necessitating more relevant studies. Moreover, current research spans multiple fields, including serum, airway epithelial cells, and bronchoalveolar lavage fluid. To determine the diagnostic and therapeutic value of EV-microRNA expression levels in different tissues, larger-scale clinical studies are needed in order to form a standardized diagnostic and therapeutic consensus.

### 3.4. EV-microRNAs and Lung Cancer

Lung cancer is the leading cause of cancer-related deaths worldwide. In 2020, there were approximately 2.2 million new cases of lung cancer globally, with about 1.8 million deaths [92]. Meanwhile, the high cost of treating lung cancer places a heavy burden on patients and the social healthcare system.

The diagnosis of lung cancer currently relies on imaging examinations, biopsy, and pathological analysis. Treatment of lung cancer varies based on the type and stage of the cancer as well as the overall health condition of the patient. Early-stage lung cancer may be curable with surgery, while the treatment for advanced lung cancer may involve chemotherapy, targeted therapy, immunotherapy, and radiotherapy [93,94]. Studies have shown that screening with low-dose CT scans in high-risk populations can improve the early diagnosis rate of lung cancer, thereby reducing the mortality rate of lung cancer [95,96]. Early diagnosis may be key to reducing the global burden of lung cancer [97].

EV-microRNAs, as biomarkers, can be detected in early-stage lung cancer through blood, sputum, and other bodily fluid samples, enabling early diagnosis. Moreover, the confirmation of lung cancer heavily depends on a biopsy and pathological examination. Liquid biopsies, including EVs, provide a possible alternative approach for patients who cannot tolerate a traumatic invasive biopsy [98]. Studying EV-microRNAs in the development of lung cancer can also contribute to better understanding the molecular mechanisms of the disease, thereby providing a basis for developing new treatment strategies. Furthermore, lung cancer patients often develop resistance to anti-tumor treatments. Research on EV-microRNAs may reveal the molecular mechanisms of drug resistance. Monitoring the expression changes of specific microRNAs can provide crucial information for clinical decision-making, aiding in the achievement of personalized treatment for lung cancer.

#### 3.4.1. EV-microRNAs as Biomarkers for Lung Cancer

Some studies investigate the differences in EV-microRNA between lung cancer patients and healthy individuals or those with other disease states, aiming to explore the potential of EV-microRNA as biomarkers for lung cancer. In 2017, Qingyun Liu et al. [99] analyzed plasma exosomal microRNA from 10 patients with lung adenocarcinoma and 10 matched healthy controls. They found that high levels of exosomal miR-23b-3p, miR-10b-5p, and miR-21-5p were independently associated with poorer overall survival rates. These microRNAs could improve the accuracy of survival prediction and potentially serve as promising non-invasive prognostic biomarkers for NSCLC. Grimolizzi et al. [100] compared the expression levels of miR-126 in the serum, exosomes, and exosome-depleted serum of patients with early and advanced NSCLC, as well as in healthy individuals. They observed that, compared to healthy individuals, miR-126 was predominantly present in exosomes in both early and advanced NSCLC patients. From these findings, they thus inferred that exosomal miR-126 might play a significant role in regulating the NSCLC microenvironment and tumor progression. Xiance Jin et al. [101] compared tumor-derived exosomal microRNAs in the plasma of patients with early-stage NSCLC and healthy individuals. The results indicated that, compared to healthy volunteers, patients with adenocarcinoma (AC) and squamous cell carcinoma (SCC) of the lung had 11 and 6 types of microRNAs with elevated expression levels, respectively, and 13 and 8 types with reduced expression levels. Among these, miR-181-5p, miR-30a-3p, miR-30e-3p, and miR-361-5p were specific to AC, while miR-10b-5p, miR-15b-5p, and miR-320b were specific to SCC. They concluded that these tumor-derived exosomal microRNAs might serve as biomarkers for the early diagnosis and differentiation of NSCLC. In 2017, Yan Wang et al. [102] investigated the differentially expressed exosomal microRNAs in malignant pleural effusions associated with lung adenocarcinoma (APE), tuberculous pleurisy effusions (TPE), and other benign pleural effusions (NPE). They observed significant abnormalities in microRNAs, with 9 microRNAs (miR-205-5p, miR-483-5p, miR-375, miR-200c-3p, miR-429, miR-200b-3p, miR-200a-3p, miR-203a-3p, and miR-141-3p) expressed at higher levels in exosomes from APE compared to TPE or NPE. Additionally, 3 microRNAs (miR-148a-3p, miR-451a, and miR-150-5p) showed differences in expression between TPE and NPE. These distinct microRNA profiles might serve as biomarkers for the differential diagnosis of pleural effusions, although further validation with larger sample sizes is needed. In 2018, Valeriy Poroyko et al. [103] compared the microRNA profiles in serum-derived exosomes from patients with small cell lung cancer (SCLC) and non-small cell lung cancer (NSCLC) to those from a healthy control group. They identified 17 microRNAs that were differentially expressed between cancer patients and healthy controls and noted significant differences in exosomal microRNA expression from patients before and after chemotherapy. Based on these findings, they inferred that exosomal microRNAs could serve as non-invasive markers for detecting lung cancer, differentiating lung cancer subtypes, and monitoring treatment responses. In 2019, Yi Zhang et al. [104] explored the expression characteristics of serum exosomal microRNAs in patients with NSCLC compared to a healthy control group. This study showed that the expression of serum exosomal miR-17-5p was significantly upregulated in NSCLC patients. They then constructed a 4-molecule diagnostic panel consisting of miR-17-5p and three tumor markers (CEA, CYFRA21-1, and SCC). The area under the ROC curve for this panel was 0.860 in the training set and 0.844 in the validation set, demonstrating high diagnostic efficacy. Jia-Tao Zhang et al. [105] investigated the relationship between plasma EV-microRNAs in patients with solitary pulmonary nodules (SPN) and the benign or malignant nature of the nodules. By constructing a support vector machine (SVM) model, they identified that miR-185-5p/miR-32-5p and miR-140-3p/let-7f-5p had the highest diagnostic value. The sensitivities and specificities were 85.1% and 75% in the training set, and 59.3% and 100% in the validation set, respectively. They concluded that EV-microRNAs might serve as potential non-invasive diagnostic tools for lung cancer. In 2021, Yuan et al. [106] confirmed through public database and clinical sample validation that there is a significant difference in the expression levels of miR-10b in plasma EVs between lung adenocarcinoma patients and the control group. They also compared the diagnostic value of plasma EV miR-10b, plasma miR-10b, and various tumor markers (AFP, CEA, CYFRA21-1, CA125, CA153, CA199, CA724, pro-gastrin-releasing peptide, and neuron-specific enolase) using ROC curves. Their findings indicated that EV-miR-10b was superior as a diagnostic marker. In 2022, Zheng et al. [107] investigated the microRNAs in circulating small extracellular vesicles (CirsEV) of up to 208 patients with indeterminate pulmonary nodules (IPN), including a training cohort of 47 cases, a test cohort of 62 cases, and an external cohort of 99 cases for validation. They constructed a model based on five differentially expressed microRNAs (let-7b-3p, miR-101-3p, miR-125b-5p, miR-150-5p, and miR-3168) to distinguish the benign and malignant nature of IPNs. The model demonstrated an AUC value of 0.920 for the ROC in the training cohort, which was confirmed in the test cohort and the external validation cohort. They further studied the differences in CirsEV-microRNAs between the benign and malignant subgroups of IPNs with a diameter ≤ 1 cm, and the results indicated that CirsEV-microRNAs can also assist in the differential diagnosis of IPNs with a diameter ≤ 1 cm.

#### 3.4.2. EV-microRNAs in the Development of Lung Cancer

Some studies focus on the role of EV-microRNA in the development and progression of lung cancer, aiming to better understand the molecular mechanisms underlying lung cancer pathogenesis and to explore new possibilities for its diagnosis and treatment. Tumor occurrence and development is a complex, multi-step process involving the dysregulation of cell growth, differentiation, and cell death. By investigating how EV-microRNAs influence these processes, researchers hope to uncover key insights into lung cancer biology and identify potential biomarkers and therapeutic targets.

Mechanisms such as abnormal cell cycle progression and inhibition of apoptosis lead to the uncontrolled growth of tumor cells [108]. In 2019, Fang et al. [37] analyzed the microRNAs in the plasma extracellular vesicles (EVs), plasma, tumor tissues, and adjacent tissues of 153 lung adenocarcinoma patients and 75 healthy controls. They found that miR-505-5p was upregulated in the plasma EVs and tumor tissues of lung adenocarcinoma patients. Using bioinformatics methods, they predicted the direct targets of miR-505-5p and verified these targets through luciferase assays and immunoblotting. Their research revealed that miR-505-5p promotes lung cancer cell proliferation and inhibits cancer cell apoptosis by targeting the tumor protein P53-regulated apoptosis-inducing protein 1 (TP53AIP1). They concluded that miR-505-5p in plasma EVs can be transferred to lung cancer cells, where it inhibits apoptosis and promotes cell proliferation by targeting TP53AIP1.

Angiogenesis is a critical step in the process of tumor occurrence and development. It involves the formation of new blood vessels to supply the nutrients and oxygen necessary for the growth and spread of tumor tissues. This process affects the interaction between tumor cells and surrounding normal tissues and provides a pathway for tumor cells to invade and metastasize to distant sites. Through angiogenesis, tumor cells can enter the bloodstream and form distant metastases [108,109]. Some studies focus on the relationship between EV-microRNA and angiogenesis. In 2016, Yi Liu et al. [38] reported that exosomal miR-21 is involved in angiogenesis in lung cancer. They observed that HBEC cells transformed by cigarette smoke extract (CSE) transmit miR-21 to normal HBEC cells via exosomes, which promotes angiogenesis in human umbilical vein endothelial cells (HUVECs). By knocking down STAT3, they were able to reduce the level of miR-21 in exosomes derived from transformed HBEC cells, thereby blocking angiogenesis. They inferred that CSE activates the IL-6/STAT3 pathway, leading to an increase in the expression level of miR-21 in bronchial epithelial cells and their secreted exosomes. This increase in miR-21 elevates the level of VEGF in recipient cells, inducing a more malignant lung tumor phenotype through enhanced angiogenesis. In 2017, Hsu et al. [39] isolated exosomes and characterized exosomal microRNAs produced by lung cancer cells under hypoxic conditions and evaluated their effects on angiogenesis and vascular permeability. They found that the level of exosomal miR-23a produced by lung cancer cells under hypoxic conditions was significantly upregulated. Using bioinformatics methods and experimental validation, they demonstrated that exosomal miR-23a directly inhibits prolyl hydroxylase 1 (PHD1) and prolyl hydroxylase 2 (PHD2), leading to an increase in hypoxia-inducible factor-1α (HIF-1α) expression in endothelial cells. This upregulation of HIF-1α increases the expression of its target genes, VEGF and PDGF-β, promoting angiogenesis. Additionally, exosomal miR-23a increases vascular permeability and the trans-endothelial migration of tumor cells by inhibiting the tight junction protein ZO-1 in vascular endothelial cells. In mouse models, inhibiting miR-23a reduced angiogenesis and tumor growth. They concluded that lung cancer cells under hypoxic conditions communicate uniquely with endothelial cells through exosomal miR-23a, which regulates the formation of tumor blood vessels. These studies on the relationship between EV-microRNA and angiogenesis offer new possibilities for the early diagnosis of lung cancer and present potential targets for anti-angiogenic therapy in lung cancer.

The pathways of tumor metastasis include direct invasion, implantation metastasis, hematogenous metastasis, and lymphatic metastasis, among others. Tumor metastasis is a crucial factor in tumor staging, guiding diagnosis and treatment, and has significant prognostic implications for cancer patients. Identifying and inhibiting tumor metastasis is essential for improving the therapeutic effects of cancer and enhancing patient prognosis. In 2023, Shimada et al. [110] compared the EV-microRNA profiles in the plasma of early-stage lung adenocarcinoma patients to those of a healthy control group, involving 24 cases in the derivation cohort, 44 cases in the validation cohort, and 6 healthy controls. They discovered that the expression level of EV-miR-30d-5p in lung adenocarcinoma patients with lymphatic metastasis was significantly downregulated compared to those without lymphatic metastasis. Furthermore, patients with high levels of EV-miR-30d-5p had better prognoses. They suggested that the level of miR-30d-5p in EVs could serve as a promising biomarker for the early detection of lymphatic metastasis in lung adenocarcinoma patients.

Epithelial-Mesenchymal Transition (EMT) is a biological process in which epithelial cells lose their original polarity and tight intercellular junctions, transforming into cells with mesenchymal characteristics. This process involves significant changes in cell morphology, adhesion properties, migratory ability, and invasiveness [111]. Studying EMT is crucial for understanding the mechanisms of cancer invasion and metastasis and provides potential targets for developing new cancer treatment strategies. In recent years, research has increasingly focused on the relationship between EV-microRNA and EMT. In 2018, Tang et al. [112] induced epithelial-mesenchymal transition (EMT) in NSCLC cells and analyzed the small RNA profiles in exosomes derived from epithelial cells (con-exosomes), mesenchymal phenotype NSCLC cells (M-exosomes), and epithelial phenotype NSCLC cells (E-exosomes). They observed significant changes in the microRNA profiles of E-exosomes and M-exosomes compared to con-exosomes after EMT. This study further suggested that M-exosomes could be internalized by epithelial cells, and specific microRNAs within them might have driven EMT, leading to the migration and invasion of recipient cells, increasing the expression of mesenchymal markers, and ultimately enhancing cancer progression. In 2019, Zhang et al. [40] conducted research on the role of exosomes derived from bone marrow-derived mesenchymal stem cells (BMSCs) under hypoxic conditions in the metastasis of lung cancer. They found that exosomes derived from hypoxic BMSCs could be internalized by lung cancer cells. Specific EV-microRNAs within the exosomes, including miR-193a-3p, miR-210-3p, and miR-5100, were implicated in promoting EMT and the invasion and metastasis of lung cancer cells by activating the STAT3 signaling pathway.

Additionally, some studies focus on the relationship between EV-microRNAs and signaling pathways. In 2019, He et al. [41] reported that exosomal miR-499a-5p in lung adenocarcinoma promotes cell proliferation, migration, and EMT through the mTORC1 signaling pathway. Their findings showed that overexpression of miR-499a-5p led to an increase in the protein levels of the downstream effectors of the mTORC1 signaling pathway, specifically p-S6K1 and p-4E-BP1, indicating activation of the mTOR pathway. Conversely, suppression of miR-499a-5p resulted in decreased levels of these proteins. Functionally, overexpression of miR-499a-5p promoted cell proliferation, migration, and EMT, while knockdown of miR-499a-5p inhibited these processes. They concluded that exosomes enriched with miR-499a-5p can enhance tumor cell proliferation, migration, and EMT via activation of the mTOR signaling pathway. In 2021, Liu et al. [42] conducted experiments using BMSCs treated with a lentivirus-mediated knockdown of miR-328-3p. These BMSCs were then cultured under normoxic or hypoxic conditions to isolate EVs. This study found that miR-328-3p was highly expressed in EVs derived from hypoxia-treated BMSCs. These EVs were shown to promote proliferation, invasion, migration, and EMT in lung cancer cells. The miR-328-3p targets the neurofibromatosis type 2 (NF2) gene, which encodes a tumor suppressor protein involved in the Hippo signaling pathway [113]. They concluded that EVs derived from hypoxia-treated BMSCs facilitate lung cancer progression by delivering miR-328-3p, which targets and inhibits the NF2 gene, thereby inhibiting the Hippo signaling pathway.

Figure 1 Hypoxia promotes angiogenesis and metastasis in lung cancer. In lung cancer, hypoxia induces elevated levels of exosomal miR-23a released by cancer cells. This miRNA downregulates PHD1 and PHD2 in vascular endothelial cells, thereby upregulating HIF-1α and VEGF expression, which promotes angiogenesis. Moreover, miR-23a suppresses the tight junction protein ZO-1 in endothelial cells via VEGF, increasing vascular permeability and facilitating hematogenous metastasis of lung cancer cells. Hypoxia also stimulates bone marrow-derived mesenchymal stem cells (BMSCs) to secrete exosomes enriched with miR-210-3p and miR-328-3p. Upon uptake by lung cancer cells, miR-210-3p targets the SOCS1 gene, activating the STAT3 pathway by suppressing SOCS1 expression, thereby enhancing VEGF production. Similarly, miR-328-3p targets the NF2 gene, promoting lung cancer cell proliferation, migration, and epithelial-mesenchymal transition (EMT) by inhibiting the Hippo signaling pathway.

#### 3.4.3. EV-microRNAs in the Treatment of Lung Cancer

Lung cancer treatment plans are varied, including surgical interventions, chemotherapy, immunotherapy, targeted therapy, anti-angiogenesis therapy, and radiation therapy. Despite these options, significant challenges persist. Traditional chemotherapy and radiation therapy can damage healthy tissues, leading to severe side effects that negatively impact patients’ quality of life. While targeted therapy and immunotherapy offer promising advances, they depend on precise biological markers for effective guidance. The discovery and validation of these markers remain a considerable challenge. Additionally, lung cancer cells often develop resistance to certain treatments [114]. Accumulating studies have focused on the role of EV-microRNA in lung cancer treatment, exploring new avenues for overcoming these challenges.

NSCLC is the most common type of lung cancer [97]. Some early-stage NSCLC cases are eligible for surgery; however, the postoperative recurrence rate significantly impacts long-term survival and the assessment of therapeutic efficacy [115]. Consequently, this issue has become a major focus of research efforts. In 2022, Han B. et al. [116] conducted a study on biomarkers for postoperative recurrence in NSCLC. They investigated the differential expression of EV-microRNAs in the tumor-draining vein (TDV) of NSCLC patients, comparing those who experienced postoperative recurrence with those who did not. This study included a screening cohort of 18 cases and a validation cohort of 70 cases. The results indicated that EV-miR-203a-3p expression was upregulated in patients with postoperative recurrence and positive lymph node metastasis. Additionally, patients with high levels of EV-miR-203a-3p had a shorter time to recur. The researchers concluded that miR-203a-3p may serve as a potential biomarker for postoperative recurrence in NSCLC.

Chemotherapy inhibits tumor development by suppressing DNA replication and cell division [117]. In 2021, Boyang Liu et al. [118] explored the role of EV-miR-433 in chemotherapy. They observed that tumor cell-derived EV-miR-433 had lower exosomal expression in the plasma of NSCLC patients with chemoresistance. Research results suggested that miR-433 inhibited the activity of the WNT/β-catenin signaling pathway by targeting TMED5, suppressed chemoresistance to cisplatin by regulating DNA damage, and inhibited tumor growth by blocking the cell cycle, promoting apoptosis, and increasing T-cell infiltration in the tumor microenvironment, providing new insights into potential prognostic biomarkers and therapeutic targets for patients with NSCLC. However, resistance to chemotherapy can develop in some patients. Research on the relationship between EV-microRNA and chemoresistance has been growing. In 2017, Wei et al. [119] conducted research on exosomal miR-222-3p derived from gemcitabine-resistant (GR) lung adenocarcinoma cells. They observed that exosomes secreted by GR lung adenocarcinoma cells were internalized by parentally sensitive lung adenocarcinoma cells via caveolin-dependent and lipid raft-dependent endocytosis mechanisms. These exosomes exhibited elevated levels of miR-222-3p, which promoted the proliferation, gemcitabine resistance, migration, invasion, and anti-apoptotic capabilities of parentally sensitive cells. Using a 3′ untranslated region reporter assay, they confirmed that miR-222-3p directly targeted SOCS3, a negative regulator of the JAK/STAT signaling pathway crucial for cell proliferation, survival, and inflammation. The researchers suggested that uptake of exosomes containing miR-222-3p from GR tumor cells by other tumor cells could downregulate SOCS3, thereby enhancing the malignant phenotype, including increased proliferation, drug resistance, migration, and invasion. Furthermore, their study revealed that NSCLC patients with higher serum levels of exosomal miR-222-3p had poorer prognoses, suggesting that exosomal miR-222-3p might serve as a biomarker for predicting response to gemcitabine therapy.

Immunotherapy is important in the treatment of advanced NSCLC, especially in patients with negative driver genes. While the expression level of PD-L1 in tumor cells can predict the tumor’s response to immune checkpoint blockers (ICB) to some extent, the definitive correlation between PD-L1 expression levels and overall survival is not always observed [114]. Therefore, to improve the precision and predictability of treatment, some studies have explored other potential biomarkers and molecular tools. In 2020, Takehito Shukuya et al. [120] collected plasma samples from NSCLC patients before PD-1/PD-L1 monotherapy and analyzed microRNAs in plasma and plasma EVs. They observed significant differences in 32 plasma microRNAs and 7 EV-microRNAs between the responders and non-responders. The researchers thus inferred that specific microRNAs in plasma and plasma EVs might serve as predictive biomarkers for response to anti-PD-1/PD-L1 therapy.

Driver genes such as EGFR (Epidermal Growth Factor Receptor) have a significant role in the development and progression of lung cancer. EGFR-TKIs (EGFR Tyrosine Kinase Inhibitors) target the active site of the tyrosine kinase domain of EGFR, inhibiting its phosphorylation and activation and slowing down the progression of the tumor [121]. In 2021, Chien-Chung Lin et al. [122] discovered that exosomes from EGFR-mutant cells could alter the sensitivity of EGFR wild-type cells to gefitinib. They observed that miR-200c was most abundant in exosomes from EGFR-mutant cells and noted an upregulation of miR-200c and miR-200a in plasma exosomes from patients who responded well to EGFR-TKIs. After transfecting miR-200c into EGFR wild-type cells, they found that the EMT pathway, a downstream signaling pathway of EGFR, was inactivated, enhancing the sensitivity of EGFR wild-type cells to gefitinib. They concluded that miR-200c in exosomes derived from EGFR-mutant cells might enhance the sensitivity of EGFR wild-type cells to gefitinib by affecting EGFR downstream signaling pathways and pathways related to EGFR-TKI resistance, such as the EMT and BIM pathways. This may explain why tumors with low-frequency EGFR mutations still respond to EGFR-TKIs in some cases [122] and provide a potential new tool for the prognosis and guidance of treatment for EGFR-mutated NSCLC. Meanwhile, some patients develop EGFR-TKIs resistance. In 2018, Yan Zhang et al. [123] discovered that exosomal microRNAs derived from gefitinib-resistant lung adenocarcinoma cells can promote drug resistance in gefitinib-sensitive lung adenocarcinoma cells. They observed that exosomes from gefitinib-resistant tumor cells could be internalized by gefitinib-sensitive tumor cells and found an upregulation of miR-214 in both the resistant cells and their derived exosomes. By applying microRNA antagonists to inhibit miR-214, they successfully reversed the drug resistance mediated by these exosomes. They inferred that gefitinib-resistant lung adenocarcinoma cells might mediate resistance through the transfer of exosomal miR-214, suggesting a potential new approach for treating gefitinib-resistant lung cancer.

Radiotherapy is important in the treatment of lung cancer [124]. With technological advances, incidental irradiation has been reduced, making radiotherapy more precise and targeted, as well as expanding its indications [125]. It is estimated that 77% of lung cancer patients have evidence-based indications for radiotherapy [126]. In 2024, Hongwen Sun et al. [127] investigated the relationship between exosomal miR-101-3p derived from BMSCs and the sensitivity to radiotherapy. Their study demonstrated that miR-101-3p promotes the expression of Retinoic Acid Induced 2 (RAI2) via suppressing the activity of Zeste enhancer homolog 2 (EZH2), inhibits DNA damage repair, and activates the PI3K/AKT/mTOR signaling pathway. This leads to the promotion of autophagy and a reduction in cell viability, ultimately enhancing the sensitivity of NSCLC to radiotherapy. However, some patients still experience a reduction in therapeutic effect, a phenomenon known as radioresistance. In 2016, Yiting Tang et al. [128] discovered that miR-208a was upregulated in the serum of patients after radiotherapy. Further in vitro experiments showed that miR-208a could be induced in lung cancer cells by X-ray radiation, transmitted through exosomes, and contribute to radio-resistance by targeting p21 and activating the AKT/mTOR pathway. In 2017, Zheng et al. [129] found that EVs secreted by lung cancer cells could enhance the proliferation and migration of human umbilical vein endothelial cells (HUVECs), with this angiogenic effect being more pronounced when the lung cancer cells were exposed to X-ray radiation. They observed a significant upregulation of EV-miR-23a during this process, which targeted Phosphatase and Tensin Homolog (PTEN). They concluded that the miR-23a/PTEN pathway plays a key role in EV-induced angiogenesis, contributing to increased radioresistance.

### 3.5. EV-microRNAs and COVID-19

Severe acute respiratory syndrome coronavirus 2 (SARS-CoV-2) was first reported in Wuhan, China, at the end of 2019 [130,131] and caused a global pandemic known as coronavirus disease 2019 (COVID-19) [7]. Since the outbreak, COVID-19 has quickly become a public health event of global concern. Rapid replication of SARS-CoV-2 in the lungs may trigger a strong immune response, leading to pneumonia, acute respiratory distress syndrome (ARDS), and respiratory failure [132]. Currently, the reverse transcription polymerase chain reaction (RT-PCR) test is the gold standard for diagnosing COVID-19, while imaging and serological tests can serve as auxiliary diagnostic methods [132,133]. Treatment includes antiviral drugs, immunomodulators, and symptomatic supportive therapy. COVID-19 vaccines are one of the main methods to prevent infection and alleviate symptoms. This study of EV-microRNA provides a new perspective for diagnosis, treatment, and prevention. Studying EV-microRNAs in COVID-19 may enhance our understanding of the virus and lead to new medical breakthroughs.

Some studies focus on the relationship between EV-microRNAs and the development of COVID-19. In 2021, Agnes S. Meidert et al. [134] investigated the differential expression of microRNAs between patients with COVID-19 Pneumonia and healthy controls. They found that in patients with COVID-19 Pneumonia, 43 microRNAs were significantly differentially expressed compared to the healthy control group. Additionally, in patients with COVID-19 ARDS, 20 microRNAs were significantly differentially expressed compared to those with COVID-19 pneumonia. From this, they inferred that microRNAs in EVs may play an important regulatory role in the pathophysiology of COVID-19. These microRNAs may serve as biomarkers for COVID-19 or potential therapeutic targets. In 2021, Yanbo Wang et al. [43] conducted a study on the inhibitory effect of EV-microRNAs on the replication of SARS-CoV-2. They observed that the expression of four microRNAs (miR-7-5p, miR-24-3p, miR-145-5p, and miR-223-3p) was significantly reduced in elderly and diabetic patients. These microRNAs were able to directly inhibit the expression of the SARS-CoV-2 spike protein and viral replication. They also discovered that exosomes in the serum of young individuals could effectively suppress the expression of the spike protein and the replication of SARS-CoV-2, but this inhibitory effect was significantly weakened in elderly and diabetic patients. Additionally, long-term exercise could increase the expression of these microRNAs in the blood of healthy volunteers and enhance their inhibitory effect on the expression of the spike protein and the replication of SARS-CoV-2. From this, they concluded that circulating microRNAs might act as endogenous antiviral molecules, directly inhibiting the replication of SARS-CoV-2. This research provides a new perspective for understanding the increased susceptibility of elderly and diabetic patients to COVID-19 infection and may offer new strategies for the clinical diagnosis and treatment of COVID-19. In 2023, Tsai-Ling Liao et al. [44] discovered that the SARS-CoV-2 spike protein can directly activate platelets and increase the levels of EV-miR-21 and let-7b. These microRNAs can interact with TLR7/8, promoting the production of reactive oxygen species (ROS) and the formation of hyperactive neutrophil extracellular traps (NETs), which then lead to the formation of immune thrombi and the deterioration of the disease [135].

Some studies focus on EV-microRNA as a therapeutic target for COVID-19. EVs derived from mesenchymal stem cells (MSCs) are considered a potential option for treating COVID-19 due to their ability to alleviate inflammatory responses and promote the repair of damaged tissues [136,137]. In 2021, Iago Carvalho Schultz et al. [138] conducted a study exploring the potential of mesenchymal stem cell (MSC)-derived EV-microRNAs as therapeutic targets for COVID-19. They analyzed four microRNA datasets and one mRNA dataset derived from different stem cell tissue sources (bone marrow, umbilical cord, and adipose tissue). They examined the EV-microRNAs present in at least two datasets and their potential target mRNAs. This study found that these EV-microRNAs target excessive cytokines and chemokines, genes that induce cell death, and the coagulation cascade. They concluded that MSC-derived EV-microRNAs might serve as a potential multi-target approach capable of improving the survival rates of patients with severe COVID-19. The PANoptosome is a protein complex involved in inflammatory cell death processes, including pyroptosis, apoptosis, and necroptosis. Its activation is associated with tissue damage, a reduction in the number of immune cells, and improper activation of the immune system. This complex may be a key factor in COVID-19 [139,140]. In 2021, Zafer Çetin et al. [140] conducted a literature review and bioinformatics analysis, revealing that the activation of the PANoptosome in COVID-19 patients correlates with disease severity. They also found that microRNAs in MSC-derived EVs could inhibit the synthesis of PANoptosome complex proteins. From this, they proposed that utilizing MSC-derived EVs as nanocarriers to deliver specific microRNAs might suppress all three pathways associated with the PANoptosome, thereby enhancing therapeutic efficacy. Additionally, plant-derived EV-microRNAs have garnered research attention as potential treatments for COVID-19. Researchers have discovered plant exosome-like nanoparticles (PELNs) in plant tissues, which possess characteristics similar to those of animal exosomes [141]. Some studies suggest that PELNs may play a role in regulating inflammatory responses [142]. In 2021, Yun Teng et al. [143] reported that miR396a-5p and other microRNAs in ginger-derived exosome-like nanoparticles (GELNs) could inhibit the expression of non-structural proteins Nsp12 and Nsp13, which are part of the RNA-dependent RNA polymerase (RdRp) in the SARS-CoV-2 virus. This inhibition might suppress the cytopathic effect (CPE) induced by SARS-CoV-2 and reduce pulmonary inflammation, offering the potential for developing new therapeutic approaches to treat COVID-19. In the same year, Sreeram Peringattu Kalarikkala et al. [144] explored the therapeutic potential of plant-derived exosome-like nanoparticles (ELNs) for COVID-19. They searched for microRNA sequences in ELNs from 13 edible plants, including grapefruit, cantaloupe, peas, coconut, blueberries, tomatoes, pears, ginger, kiwi, oranges, and soybeans. Using computational prediction methods, they identified microRNAs that might target SARS-CoV-2. Among these plants, they discovered 22 microRNAs that could potentially target the SARS-CoV-2 genome. As validation, they isolated ELNs from ginger and grapefruit and confirmed the expression of some microRNAs in these ELNs that target SARS-CoV-2. These studies illustrate the potential of plant-derived microRNAs in cross-species regulation and the possibility of ELNs as a treatment against COVID-19.

Some studies explore the relationship between EV-microRNA and COVID-19 vaccines. Currently, these vaccines include mRNA vaccines, adenovirus vector vaccines, protein subunit vaccines, and inactivated vaccines [145]. The antigens in these vaccines are recognized by antigen-presenting cells (APCs), which activate T cells and produce memory T cells. Cytokines from Th1 cells activate CD8+ T cells, leading to the production of perforin and the death of infected cells. Th2 cells activate B cells to produce plasma cells and memory B cells. When the antigen reinfects the body, memory T cells and memory B cells rapidly proliferate and differentiate, mediating a quick cellular and humoral immune response. This response reduces the severity of the COVID-19 infection [145]. As a potential method to prevent COVID-19 infection, alleviate symptoms, and reduce transmission, the safety and efficacy of vaccines have garnered significant attention from both the public and scholars [146,147]. mRNA vaccines have been widely used during the COVID-19 pandemic due to their rapid development, cell-free production, high immunogenicity, flexibility, and scalability. However, they can also cause side effects, ranging from mild reactogenicity to rare, severe conditions [148]. Balancing the immunogenicity and reactogenicity of vaccines remains an urgent issue to address [149]. In 2022, Yusuke Miyashita et al. [150] analyzed the levels of microRNA in serum extracellular vesicles (EVs), pro-inflammatory cytokines in serum, and antibody titers after vaccination with the COVID-19 mRNA vaccine (BNT162b2). They found that the level of miR-92a-2-5p in EVs was negatively correlated with the severity of adverse reactions, while the level of miR-148a was associated with specific antibody titers. They concluded that circulating EV microRNAs have the potential to serve as biomarkers for assessing vaccine efficacy and predicting adverse reactions.

## 4. Conclusions

Respiratory diseases are intricately interconnected. Firstly, some respiratory diseases share common risk factors; for instance, smoking is a significant risk factor for both COPD and lung cancer, suggesting that their pathogenesis may involve overlapping molecular pathways. EV-microRNAs can serve as a unique angle for delving deeper into this issue. As previously mentioned, miR-210 [31,40], miR-21 [32,38], and miR-126 [34,100] are found to exhibit altered expression levels in both COPD and lung cancer. Further investigation into their roles in the onset of these diseases may enhance our understanding of pathogenic mechanisms. Secondly, some respiratory diseases coexist and influence each other, such as Asthma-COPD Overlap Syndrome (ACOS). Patients diagnosed with ACOS typically experience symptoms more frequently and have a more severe decline in lung function compared to those with asthma or COPD alone, and there is currently no evidence-based best treatment approach [151]. EV-microRNAs, as potential molecular links in the complex network of ACOS pathogenesis, may provide insights into the mechanisms of ACOS development, thereby advancing the development of treatments such as targeted therapy. Lastly, as one of the key molecules that regulate inflammatory responses, cell proliferation, apoptosis, and angiogenesis, EV-microRNAs can act as a bridge in respiratory diseases, aiding in the exploration of better diagnostic and therapeutic methods for the coexistence of multiple respiratory diseases. In essence, as a crucial player in intercellular communication, EV-microRNAs participate in various molecular mechanisms and play an important role in the complex network of respiratory diseases [152]. Investigating their role deepens our understanding of respiratory diseases and offers possibilities for the development of new diagnostic and therapeutic approaches.

Screening and early diagnosis of cancer can greatly improve prognosis, making it a focal point in research. Liquid biopsy has great potential in the diagnosis of cancer, especially in early detection. Liquid biopsy detects and monitors diseases by analyzing biomarkers in blood or other body fluids. Compared to traditional biopsy methods, liquid biopsy has many advantages, including non-invasiveness, repeatability, and real-time monitoring. EV-microRNAs, as important molecules that reflect the state of the disease, have always been one of the research focuses of liquid biopsy [153,154,155,156,157,158]. As previously mentioned, some EV-microRNAs have differential expression levels in lung cancer patients, providing more possibilities for the liquid biopsy of lung cancer. In addition, EV-microRNAs are involved in the occurrence, development, treatment, and drug resistance of lung cancer, offering possibilities for real-time monitoring of lung cancer treatment effects and guiding treatment plans.

The current technology for quantifying EV-microRNA is generally reliable, with high sensitivity and accuracy. However, the widespread distribution of EVs in complex body fluids, their small size, and their low density make it challenging to effectively isolate EVs from samples. Extraction methods often struggle to ensure both the purity and yield of EVs. Meanwhile, cost-effectiveness is an unsolved problem before laboratory techniques can be applied on a clinical scale. It should also be pointed out that, while numerous studies have highlighted distinct characteristics of EV-microRNAs in specific disease states, there is currently no consensus on which EV-microRNAs in which samples can effectively serve as diagnostic markers for specific diseases or as markers for assessing efficacy and predicting prognosis. Extensive clinical studies are essential to validate the accuracy and reliability of EV-microRNAs as biomarkers, aiming to identify the optimal combination of EV-microRNAs for biomarker applications.

Despite the accumulating research on microRNA as a therapeutic target, the development of microRNA-based drugs encounters several challenges, such as precise targeting, stability, delivery efficiency, and potential immunogenicity [13]. For EV-microRNA to be applied in the treatment of respiratory diseases, it is crucial to ensure that the therapeutic agent can effectively bind to key gene sites involved in the disease process and accurately regulate the expression of specific RNA molecules. Additionally, ensuring the safety of these agents is essential to preventing serious adverse reactions. Furthermore, to minimize the risk of the immune system clearing therapeutic EVs, utilizing autologous EVs as carriers for delivering therapeutic molecules is recommended [159]. Additionally, enhancing the binding efficiency of EVs to specific target tissues may require surface modifications to incorporate molecules with affinity for those tissues. Addressing these requirements demands extensive research, rigorous clinical trials, and ongoing exploration and development of EV-related technologies.

In recent years, some researchers have endeavored to develop microRNA delivery therapies based on EVs or nanomaterials. This innovative treatment method has been studied in relation to diseases such as asthma [91] and lung cancer [160]. However, this technology is still in exploratory stages, since targeted delivery to specific types of cells remains challenging [152].

In summary, EV-microRNAs play a crucial role in the pathogenesis of COPD, asthma, lung cancer, and COVID-19 by affecting key targets and pathways through intercellular communication. These microRNAs hold promise as potential biomarkers and therapeutic targets, with the potential to improve the diagnosis and treatment of these diseases and advance the application of precision medicine in clinical practice. However, current research on EV-microRNAs in respiratory diseases is mainly focused on basic research. Significant progress is still needed before EV-microRNAs can be clinically applied as biomarkers or therapeutic targets.

## Figures and Tables

**Figure 1 ijms-25-09147-f001:**
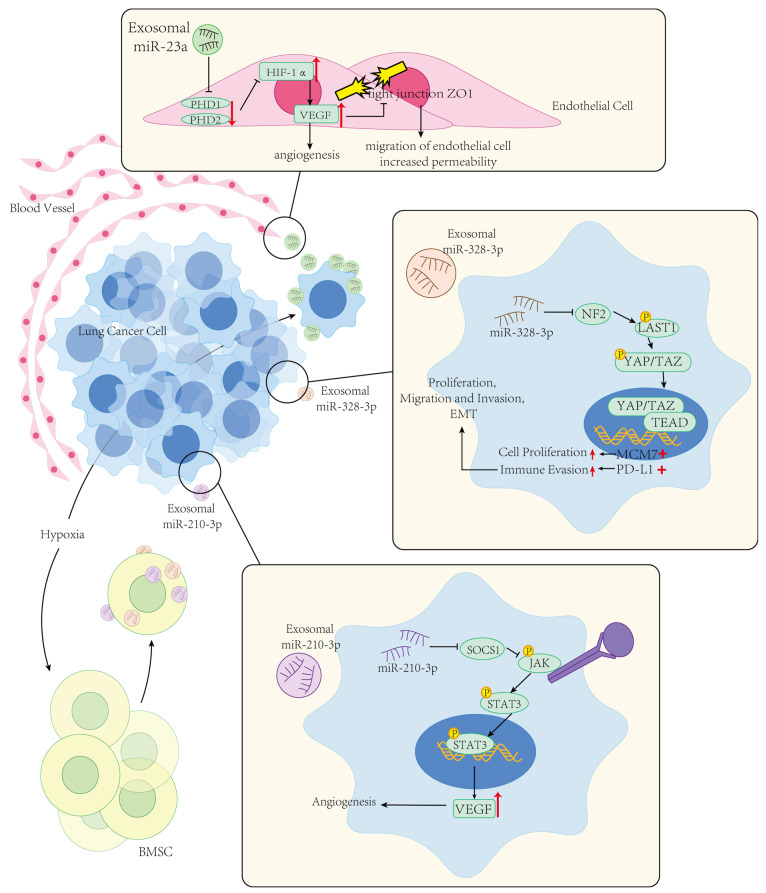
EV-miRNAs in the Angiogenesis and Metastasis of Lung Cancer.

**Table 1 ijms-25-09147-t001:** The biogenesis and possible function of EVs.

Vesicle	Size(nm)	Biogenesis	Possible Function
Exosome	30–150	endocytosis→cargo sorting→MVB maturation and secretion	Cellular interactions and signaling
Microvesicle	100–1000	plasma membrane budding and shedding	Cellular interactions and signaling
Apoptotic body	50–5000	chromatin and nucleus condense in cell apoptosis	Clearance of apoptotic debris

**Table 2 ijms-25-09147-t002:** Isolation techniques of EVs.

Isolation Technique	Principle	Advantage	Disadvantage
Ultracentrifugation	High-speed centrifugal force to sediment EVs	Suitable for large volumes and various samples; yields relatively pure EVs	Time-consuming; complex procedures; impact on EV activity
Size-based separation	Physical barriers (chromatography, filtration, etc.)	Efficient; easy to perform; little impact on EV activity	Relatively low purity; limited sample volume
Immunoaffinity Capture	Antibody-based binding of specific EV markers	High specificity; high purity	Expensive; limited to specific EV subtypes
Polymer Precipitation	Use of polymers to alter the solubility of EVs	Efficient; easy to perform	May co-precipitate non-EV components; relatively low purity
Microfluidics	Manipulation of fluids at microscale through microchannel of microfluidic chip	High precision; integrates multiple steps; efficient; little impact on EV activity	Requires specialized equipment; limited throughput
Kit Method	Commercially available kits for EV isolation	User-friendly	Often expensive; variable purity and yield

**Table 3 ijms-25-09147-t003:** Quantification methods of miRNA.

Method	Principle	Advantage	Disadvantage
Northern blot	Separation of specific-sized RNA fragments by electrophoresis, followed by membrane transfer and probe hybridization detection	Reliable results without amplification; can detect both miRNAs and their precursors	Low sensitivity; time-consuming; requires large number of RNA samples; RNA is prone to degradation
Microarray	Hybridization of labeled miRNAs with DNA probes fixed on a solid surface	High throughput; can simultaneously detect multiple samples	High cost; complex operation; sensitivity and specificity are limited; requires standardized probe sets
qRT-PCR	Reverse transcription of miRNA into cDNA; followed by real-time monitoring with fluorescently labeled probes and primers	Gold standard for miRNA detection; highly sensitive and accurate; suitable for quantitative analysis	Complex primer design; requires consideration of RNA integrity; cDNA synthesis; primer design; and other parameters
NGS	High-throughput sequencing analysis of multiple small RNA fragments	No need for whole genome sequence; extensive coverage of miRNAs at different expression levels	Complex data processing; relatively high cost

**Table 4 ijms-25-09147-t004:** EV-miRNAs in the Pathogenesis of Respiratory Diseases.

	miRNAs	Samples	Target	Expression	Function	Pathogenesis	References
COPD	miR210	HBEC	ATG7	upregulated	Control autophagy process	Promote airway remodeling	[31]
	miR21	HBEC	VHL	upregulated	Promote myoblast differentiation	Promote airway remodeling	[32]
	miR93	Lung tissue	DUSP2	upregulated	Activate JNK pathway in macrophages	Contribute to emphysema	[33]
	Let-7d, miR-191, miR-126, miR-125a	plasma		upregulated	Compromise the phagocytic function of macrophages	Escalate endothelial damage and an inflammatory response	[34]
Asthma	miR-1470	MSC	P27KIP1	upregulated	Induce CD4+CD25+FOXP3+ Tregs differentiation	Regulate inflammatory response	[35]
	miR-370	Lung tissue	FGF1	downregulated	Suppress the FGF1/MAPK/STAT1 axis	Reduce inflammation, inhibit airway remodeling	[36]
Lung Cancer	miR-505-5p	Plasma and tumor tissue	TP53AIP1	upregulated	Inhibits cell apoptosis and promotes cell proliferation	Contribute to an abnormal cell cycle	[37]
	miR-21	HBEC	VEGF	upregulated	Form new blood vessels	Contribute to angiogenesis	[38]
	miR-23a	lung cancer cells	HIF-1α	upregulated	Form new blood vessels	Contribute to angiogenesis	[39]
	miR-193a-3p, miR-210-3p, miR-5100	Plasma and Lung tissue	STAT3	upregulated	Activate STAT3 signaling-induced EMT	Promote metastasis of cancer cells	[40]
	miR-499a-5p	Lung tumor cells	mTORC1	upregulated	Activate the mTOR signaling pathway	Promote cell proliferation, migration, and EMT	[41]
	miR-328-3p	BMSC	NF2	upregulated	Inhibit the Hippo signaling pathway	Promote proliferation, invasion, migration, and EMT	[42]
COVID19	miR-7-5p, miR-24-3p, miR-145-5p, miR-223-3p	plasma	S protein	downregulated	Inhibit S protein expression and SARS-CoV-2 replication	Act as endogenous antiviral molecules	[43]
	miR-21, let-7b	plasma	TLR7/8	upregulated	Promote the production of ROS and the formation of NETs	Form immune thrombi	[44]

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
