# Peer review of "Extracellular Vesicle microRNA: A Promising Biomarker and Therapeutic Target for Respiratory Diseases"

_ijms, 2024, doi:10.3390/ijms25179147_

Round 1

Reviewer 1 Report

Comments and Suggestions for Authors

The topic of paper is of great interest. The current review highlights key findings that show the importance of miRNAs to the regulation of cells that are found in the lung and their contribution to the mechanisms underpinning respiratory disease.

A review is a synopsis that summarizes research that gives an idea of the current  state of the issue to be analysed. Also, in the review, a critical assessment of other research pn a given topic is carried out, a process that helps us put the topic on its context. 

One of the aspects that characterizes the integrative review is that a series of  criteria are established in a transparent manner that ensure the quality of the information ot the review. Was any quality evalaution instrument applied to the selected papers? 

There are some of the aspects included in this paper that are excessively general, especially those that are related to each of the diseases that are described. Tge review is too long and can be shortened. 

Why do the authors focus on the diseases and not others? This issue must be explained.

A summary table could be included with EV-microRNA roles in the respiratory diseases. 

The conclussion section is poor. The conclusions are excessively broad and generic. More recent and novel information found in the review must be included.

Author Response

Comment 1:

A review is a synopsis that summarizes research that gives an idea of the current state of the issue to be analysed. Also, in the review, a critical assessment of other research on a given topic is carried out, a process that helps us put the topic on its context.

One of the aspects that characterizes the integrative review is that a series of criteria are established in a transparent manner that ensure the quality of the information ot the review. Was any quality evalaution instrument applied to the selected papers?

Response 1:

We appreciate your insightful comment regarding the importance of quality evaluation in review article. We fully agree that a transparent and systematic approach is crucial for ensuring the integrity and reliability of the information presented.

Given the diverse nature of the studies cited in our review—including clinical, animal, and in vitro studies—we found it challenging to apply a single, established evaluation instrument suitable for all study types. Therefore, we developed a qualitative evaluation method that considers the common characteristics of these different study types.

Our evaluation focused on three primary aspects:

  1. Study Design: This includes the clarity of the research objectives and questions, the suitability of the study design (observational or experimental), the appropriateness of control groups, and whether there was follow-up verification of the results.
  2. Sample Selection and Sample Size: We evaluated the representativeness of the sample, potential biases, the adequacy of the sample size for ensuring statistical reliability, and the accuracy of data collection methods.
  3. Statistical Methods and Results Interpretation: This involved assessing the data cleaning process, the suitability of statistical methods for the data type and research design, and the interpretation of the results, ensuring that there was no over-interpretation or overlooking of potential limitations.

Based on these evaluations, each study was categorized into "High," "Medium," or "Low" overall quality. Specifically, a study was classified as "High" overall quality if no less than two of the three aspects were rated as "High," and as "Low" overall quality if no less than two aspects were rated as "Low."

The studies we assessed are highlighted in the resubmitted manuscript.

To present these results clearly, we used an affiliated Excel file, which provides the relevant information and quality assessment for each study. This file has been included as supplementary material in the revised manuscript.

We believe this approach, while qualitative, provides a relatively comprehensive assessment of the studies included in our review. We appreciate your suggestion, which has prompted us to clarify and expand upon our methodology.

Comment 2:

       There are some of the aspects included in this paper that are excessively general, especially those that are related to each of the diseases that are described. The review is too long and can be shortened.

Why do the authors focus on the diseases and not others? This issue must be explained.

Response 2:

       Thank you for your valuable feedback regarding the scope and focus of the review.

Regarding the Length and General Aspects:

We acknowledge your concern about the length of the manuscript and the excessiveness of certain sections. In response, we have revised the introduction of each disease section to eliminate redundant content, particularly regarding overly simplistic or widely known explanations of etiology and mechanisms. Instead, we have emphasized the epidemiology and the current status of diagnosis and treatment for each disease. This approach highlights the diagnostic and therapeutic dilemmas faced in managing these conditions, providing a clearer rationale for our selection of diseases in the review.

Regarding the Focus on Specific Diseases:

The selection of diseases included in this review was based on several criteria:

  1. Prevalence and Impact: The diseases (COPD, asthma, lung cancer, COVID19) we focused on are among the most prevalent and impactful respiratory conditions globally, with significant morbidity and mortality rates, as we’ve highlighted in the introduction of each disease of our revised manuscript.
  2. Relevance to microRNA Research: We selected diseases for which there is substantial evidence or emerging interest in the role of extracellular vesicle (EV)-associated microRNAs as biomarkers or therapeutic targets, aligning with the central theme of our review.
  3. Availability of Quality Research: We prioritized diseases for which a robust body of research exists, enabling a meaningful synthesis of findings and identification of trends.

We also added paragraph to explain the reason we focused on these four diseases in the revised manuscript under the section “1. Introduction”.

We believe that these revisions address your concerns and enhance the clarity and focus of the manuscript.

Comment 3:

       A summary table could be included with EV-microRNA roles in the respiratory diseases.

Response 3:

       We appreciate your suggestion to include a summary table highlighting the roles of EV-associated microRNAs in respiratory diseases. In response, we have made a table detailing the roles of EV-microRNAs in the pathogenesis of the diseases discussed in our review. This table has been placed before the conclusion section to provide a clear and concise overview of the relevant findings.

We believe that this addition enhances the readability of the manuscript and offers resource for understanding the specific contributions of EV-microRNAs to the pathogenesis of the diseases covered.

Thank you for the helpful suggestion, which has improved the overall quality of our review.

Comment 4:

       The conclusion section is poor. The conclusions are excessively broad and generic. More recent and novel information found in the review must be included.

Response 4:

       We appreciate your constructive feedback regarding the conclusion section of our review. In response, we have made significant revisions to enhance the depth and specificity of our conclusions.

We have condensed sections that were overly general or redundant to improve clarity. Additionally, we have summarized the previous text and incorporated discussion of the potential value of EV-miRNAs in respiratory diseases with common etiologies, coexisting respiratory diseases, and interrelated respiratory diseases. This addition provides a more comprehensive perspective on the topic.

We have also included recent research on EV-miRNAs in liquid biopsies for oncological diseases, as an example for their value as biomarkers.

We believe these updates can significantly strengthen the conclusion section. Thank you for your valuable suggestions.

Reviewer 2 Report

Comments and Suggestions for Authors

In this interesting manuscript, authors provide a broad overview of the benefits of studying Extracellular Vesicles microRNA for many different purposes. In fact EV-miRNAs have a profound impact on the onset, progression, and therapeutic responses of respiratory diseases, such as COPD,

asthma, lung cancer, and novel coronaviruses, with great potential in disease management.

EV-miRNAs can be considered as biomarkers and therapeutic targets in the diagnosis and treatment of these respiratory diseases.

Exosomes are produced by almost all human cells and correlate with an individual's genetic background and disease state, thereby informing personalized medicine and precision therapy.

Lastly their usefulness has been discussed also for their important regulatory role in the pathophysiology of COVID-19. These microRNAs may serve as biomarkers for COVID-19.

This manuscript is well written and covers different areas of literature described in a fluid and flowing way.  

Exosomes are a very topical subject to which many researchers are turning their attention. The paper discusses several aspects of lung pathology and is very useful to the reader, providing a more comprehensive picture of the current state of research on the subject.

The only aspect that I would improve is the quality of the figures which are very elementary and also of low quality

Otherwise I have nothing to observe

Comments on the Quality of English Language

the english is quite good

just some minor revisions

Author Response

Comment 1: 

The only aspect that I would improve is the quality of the figures which are very elementary and also of low quality.

Response 1:

In response to your suggestion for improvement, we have made significant revisions to enhance the quality of the figures. Specifically, we revised Figure 1 by removing illogical arrows and boxes and incorporating images of tumor cells and blood vessels to create a more visually engaging representation. We removed Figure 2, as one of the studies cited was retracted during our literature reorganization, leaving the figure incomplete.

Additionally, we have made revisions throughout the manuscript to eliminate repetitive or redundant phrases, ensuring the text is more concise and accurate.

Thank you for your valuable insights, which have helped us enhance the quality of our manuscript.

Round 2

Reviewer 1 Report

Comments and Suggestions for Authors

The work has been improved by including the suggestions.

The paper can be published in present form.